# Complex small-world regulatory networks emerge from the 3D organisation of the human genome

C. A. Brackley [1], N. Gilbert [2], D. Michieletto[1,2], A. Papantonis[3], M. C. F. Pereira[1], P. R. Cook [4] & D. Marenduzzo [1✉]

The discovery that overexpressing one or a few critical transcription factors can switch cell state suggests that gene regulatory networks are relatively simple. In contrast, genome-wide association studies (GWAS) point to complex phenotypes being determined by hundreds of loci that rarely encode transcription factors and which individually have small effects. Here, we use computer simulations and a simple fitting-free polymer model of chromosomes to show that spatial correlations arising from 3D genome organisation naturally lead to stochastic and bursty transcription as well as complex small-world regulatory networks (where the transcriptional activity of each genomic region subtly affects almost all others). These effects require factors to be present at sub-saturating levels; increasing levels dramatically simplifies networks as more transcription units are pressed into use. Consequently, results from GWAS can be reconciled with those involving overexpression. We apply this pan-genomic model to predict patterns of transcriptional activity in whole human chromosomes, and, as an example, the effects of the deletion causing the diGeorge syndrome.

[1] SUPA, School of Physics and Astronomy, University of Edinburgh, Peter Guthrie Tait Road, Edinburgh EH9 3FD, UK. [2] MRC Human Genetics Unit, MRC Institute of Genetics & Molecular Medicine, University of Edinburgh, Western General Hospital, Edinburgh EH4 2XU, UK. [3] Institute of Pathology, University Medical Center, Georg-August University of Göttingen, 37075 Göttingen, Germany. [4] Sir William Dunn School of Pathology, University of Oxford, South Parks Road, Oxford OX1 3RE, UK. ✉email: dmarendu@ph.ed.ac.uk

Transcription—the copying of DNA into RNA—is tightly regulated. Early insights into regulatory mechanisms came from work on binary on/off genetic switches controlled by one or just a few transcription factors such as the lambda and lac repressor in *Escherichia coli*[1]. Similar regulatory mechanisms are present in eukaryotes, albeit with additional complexity. For instance, a fibroblast cell can be reprogrammed into a muscle cell by a single master regulator (MYOD)[2,3] or into pluripotent stem cells by four Yamanaka factors (Oct4, Sox2, c-Myc, Klf4)[4].

Genome-wide association studies (GWAS) lead to quite a different view: gene regulation is widely distributed and involves interactions between hundreds (perhaps thousands) of loci scattered around the genome[5,6]. GWAS allow quantitative trait loci (QTLs) affecting any measurable genetic trait to be ranked in an unbiased way. With complex traits like human height, and diseases such as schizophrenia and type II diabetes, the top ten QTLs in the rank order combine to yield only modest effects, while the top one-hundred still account for less than half of the total genetic effect. Hundred more QTLs are expected to be identified as sample sizes and data resolution improve[5–7]. Expression QTLs (eQTLs) are QTLs affecting transcription of other DNA regions. Perhaps surprisingly, these are rarely found in genes encoding transcription factors or other proteins; instead, they usually involve single-nucleotide changes in non-coding elements that bind transcription factors such as active enhancers and promoters[8–10].

Results from GWAS lead to the view that most gene-regulatory networks are incredibly complex, with the activity of a given gene being affected by a panoply of eQTLs, each having a tiny effect. This is captured by the "omnigenic" model, which is based on a set of gene-interaction equations[5,6] such that the activity of almost any gene affects that of almost every other one. This model provides a useful and appealing framework to view GWAS results. However, it is difficult to compare its outputs with experimental data because it contains many parameters that are currently unknown and require fitting to training datasets.

In general, existing models for gene regulation traditionally assume post-transcriptional and biochemically mediated interactions between different genes[11,12], and disregard the role of three-dimensional (3D) chromatin structure. Here we propose an alternative but complementary framework that links transcriptional regulation directly to 3D genome structure, deliberately neglecting downstream biochemical regulation to enable unambiguous interpretation of our results. This framework is motivated by experiments showing that chromatin folding can lead to contacts between enhancers and promoters affecting transcription, and that 3D structure changes in disease[13,14]. Additionally, because our modelling is essentially fitting-free, its output can be directly compared to experiments. When the agreement is good, our model is validated; when poor, it points to some missing ingredient (such as biochemical feedback) that could be included in future models.

We use stochastic computer simulations of a polymer model for chromosome organization, in which a chain of beads represents a chromatin fibre, and a set of spheres complexes of transcription factors and RNA polymerases—which we will call "TFs" for short. Some chromatin beads are identified as transcription units (TUs), and we call them TU beads. They contain binding sites for TFs, and can be sites of transcriptional initiation (we do not discriminate between genic and non-genic promoters). As a simple starting point we only consider one type of TF that binds specifically and multivalently to TU beads, and non-specifically (i.e., with weak affinity) to every other bead. We perform 3D Brownian dynamics simulations that evolve the diffusive dynamics of the chain and associated factors. We previously showed that similar polymer models yield structures resembling those seen using chromosome–conformation–capture (3C)[15–19]

and microscopy[20]. Here, we link 3D structure to expression and transcriptional dynamics by measuring how often a TU bead is transcribed—which we do by computing the fraction of time it binds a TF. To establish the methodology, we model a 3 Mbp chromatin fragment, before going on to simulate whole human chromosomes.

Our simulations capture many features of eukaryotic regulation. For example, transcription is stochastic and bursty (in agreement with single-cell transcriptomics data), and the predicted pattern of transcriptional activity in human chromosomes correlates significantly with that observed experimentally. We also find that small-world (percolating) networks that encapsulate much of the rich complexity observed in GWAS emerge through spatial effects alone. In other words, the activity of most (probably all) TUs in our model is affected by the activity of most (probably all) other segments in the genome. We find such pan-genomic regulation critically requires non-saturating concentrations of TFs —as normally found in vivo—and that increasing concentrations dramatically simplifies the networks. This enables us to reconcile the GWAS-based view that regulatory networks are complicated with the observation that overexpressing one or a few TFs can decisively alter cell state.

## Results

We first consider a simple system where a 3 Mbp chromatin fragment is represented by a chain of 1000 beads (each 30 nm in diameter, and corresponding to 3 kbp). We select at random $N = 39$ beads and identify them as TUs (Fig. 1a; see "Methods" and Supplementary Note 1 for more details). The linear density of TUs in the fragment is similar to that in human chromosome 22. Additionally, $n$ spheres (also 30 nm in diameter) represent TFs (recall these are complexes of transcription factors and RNA polymerase II). TFs bind reversibly to TUs via a strong attractive interaction, and to all other beads weakly and non-specifically. An important feature is that TFs switch between active (binding) and inactive (non-binding) state at rate $\alpha$. Many factors switch like this in vivo (e.g., due to phosphorylation and de-phosphorylation), and switching is required to account for the rapid exchange of factors and polymerases between bound and free states seen in live-cell photobleaching experiments[21]. As ~7 out of 8 polymerases attempting to initiate at promoters dissociate with a half-life of ~2.4 s[22], our complexes generally behave like those in vivo.

While our results refer to a single patterning of TUs along the fibre, they are representative of any arbitrary random positioning of TUs: in other words the qualitative trends we present below are robust and do not depend on the particular choice of the 1D pattern of TUs along the fibre in any way.

We say a TU bead is transcribed whenever a TF lies close to it (see "Methods"), and the transcriptional activity of a TU is then the fraction of time it is transcribed during a simulation. To reflect the situation in mammalian cells (Supplementary Note 3 and ref. [23]), we typically assume there are fewer TFs than TU beads (i.e., $n = 10$ TFs in the active binding state at any time, compared to 39 TUs).

By interrogating TF-chromatin interactions at regular time intervals over hundreds of simulations, we build up a population picture of transcription. A typical configuration of the 3 Mbp fragment is shown in Fig. 1B. Strikingly, bound TFs spontaneously cluster, despite there being no attractive interactions between TUs or between TFs. Such clustering is driven by the "bridging-induced attraction"[16,24,25] that arises due to a positive feedback: when a TF forms a molecular bridge between two chromatin regions and forms a loop, the local chromatin concentration increases, making further TF binding more likely. Clusters then grow until limited by entropic costs of crowding

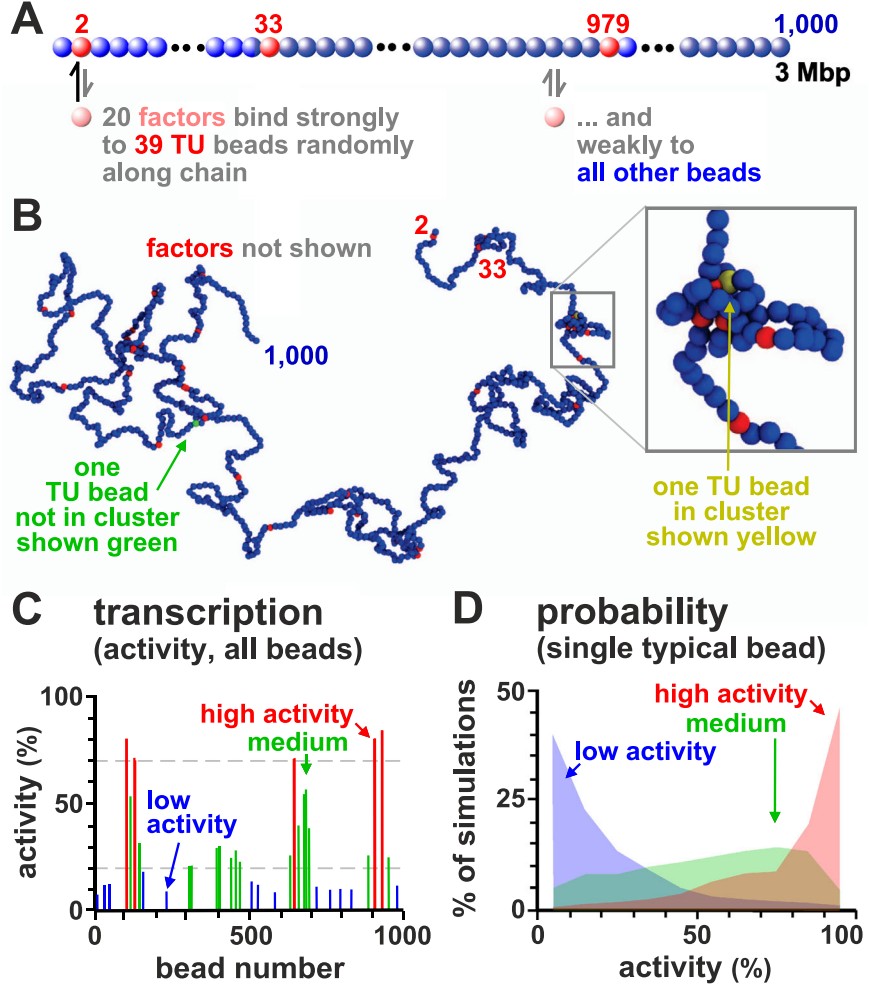

**Fig. 1 Patterns of transcriptional activity. A** Schematic of the model. Twenty TFs (pink) that switch between on/off states at rate $\alpha = 10^{-5}\tau_B^{-1}$ (with $\tau_B$ the Brownian time, see "Methods") or $0.001\,\mathrm{s}^{-1}$ bind specifically to 39 TUs (red beads) randomly positioned along the chain, and non-specifically to other beads (blue). A TU is considered transcriptionally active if associated with a TF. **B** Example conformation (TFs not shown). Some beads cluster and form loops; one TU not in a cluster (and not transcribed) is green, and another that is in a cluster (and transcribed) is yellow. Inset: zoom of boxed region. **C** Transcriptional activity for each TU bead averaged over 1000 simulations (each lasting $10^5\tau_B$). TUs are grouped according to activity, with red, green, and blue bars showing high (>70%), medium (20–70%) and low (<20%) activity, respectively. This gives a population-level measure of activity. **D** Variation of activity across simulations (reflecting cell-to-cell variation) for three representative TUs with high (red), medium (green), or low (blue) average activity (defined as in **C**).

(Fig. S1A). Most of the non-trivial phenomena described below result from such clustering. Clustering requires TF multivalency, as monovalent factors do not cluster[24]. However, the assumption of multivalency, which is common in the polymer physics literature[15], is well-founded. Several TFs are known to be bivalent or multivalent[26], and, more importantly, our spheres represent complexes of TFs and polymerases, so they will behave as multivalent binders even when the individual TFs in the complex are monovalent. Although clustering does not require any interactions between TFs, adding a weak attraction between them, as might arise for instance due to macromolecular crowding or electrostatic interactions between intrinsically disordered regions, should not qualitatively change any of the results discussed here (at least as long as TFs still microphase separate into clusters rather than undergoing macroscopic phase separation).

The clusters we observe, and which emerge through the bridging-induced attraction, are qualitatively similar to those seen in vivo, which are variously described as transcriptional compartments, hubs, super-enhancer (SE) clusters, phase-separated droplets/condensates, and factories[7,10,27–29]. They are also similar to the contact domains seen in microC[30], which are formed by

accessible DNA sites clustering together in 3D space. Clustering arising through the bridging-induced attraction has recently been found in vitro for systems of DNA and cohesin (which binds multivalently to DNA)[31].

**Transcriptional activity varies along the chromatin fibre and is highly stochastic.** As TFs have the same affinity for all TUs, one might expect each TU to be bound with equal likelihood; however, transcriptional activity (the fraction of time a TU is transcribed) varies from ~10–90% (Fig. 1C). What causes this variation? As TF copy number is limiting, and as bound TFs cluster, most transcription occurs in clusters—as is the case in vivo[7,32–34]. Since TUs are positioned irregularly along the fragment, some have closer neighbours in 1D sequence space than others, and these are inevitably the ones most likely to cluster and be transcribed. Instead, those far from their neighbours are less likely to cluster and are less active. Accordingly, the transcriptional activity of a TU anticorrelates with distance to the nearest TU along the fibre (Fig. S1B; the Spearman correlation is $r \simeq -0.94$, $p$ value $p < 10^{-12}$).

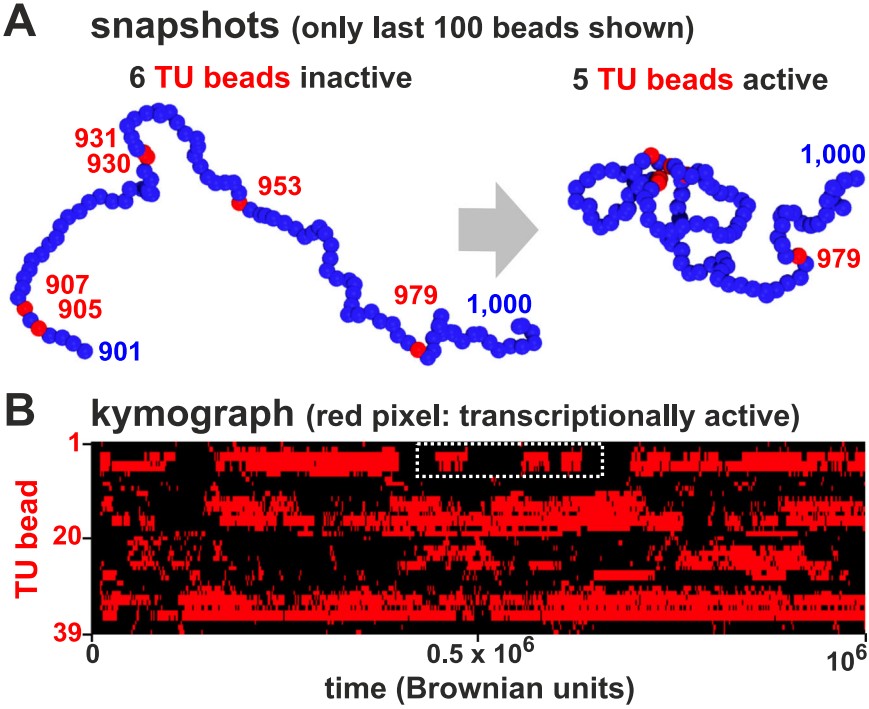

**Fig. 2 Transcriptional bursting. A** Snapshots showing a 100-bead section of the simulated chain taken at different times. Initially, none of the 5 TUs (red) are in clusters and they are all inactive; later, 4 TUs join a cluster and are close to TFs—and so are transcribed. **B** Kymograph where each row shows the changing transcription state of one TU during a simulation; pixels are colored red if the bead is associated with a TF and so transcribed, or black otherwise. White rectangle: example of bursts.

While Fig. 1C pertains to population averages of 1000 simulations, it is informative to consider each simulation independently (as in single-cell transcriptomics). Such analysis shows that transcriptional activity is stochastic, varying substantially from simulation to simulation: a TU active in some simulations may be silent in others (Fig. 1D).

**Transcriptional bursting**. During a simulation, chromatin conformation can change dramatically (Fig. 2A). Such changes often yield transcriptional "bursts"—periods of continued activity followed by silent periods (Fig. 2B)—as TUs with intermediate levels of activity repeatedly join a cluster to give a burst and then dissociate. Notably, TUs lying close to each other in sequence space often start and stop bursts coordinately due to the intrinsic positive feedback in the system (Fig. S1A).

These results are consistent with experimental observations: single cell Hi-C[35] and transcriptomics[36] show that the structure and function of each individual cell is unique, and bursting is well documented[37–40] with nearby promoters often firing together[38].

**Local chromatin architecture creates small-world percolating transcription networks**. To investigate correlations between transcriptional activities of different TUs, we compute the Pearson correlation matrix between the activities of all possible TU pairs, and identify an emergent regulatory network in which TUs form nodes (Figs. 3A and S2). Specifically, we draw an edge between two TUs whenever there is a statistically significant positive or negative correlation between their transcriptional dynamics (Fig. 3A). This network arises only due to spatial interactions, as we assume no underlying biochemical regulation.

The network shows a striking property. With $n = 10$ active TFs, most nodes are connected (Fig. 3Aii), and the fraction of TUs participating in the largest connected component is close to 1 (Fig. 3B). Such a network is said to be "percolating", which

means that any two nodes are connected by a path along edges. Our percolating networks are also "small-world", which means that most nodes can be reached from every other node by a small number of steps[41]—we provide quantitative measurements of the small world-ness of our networks in the SI (Supplementary Note 4). The small-world phenomenology is consistent with the multitude of small-effect eQTLs detected by GWAS[5,6]. Notably, the regulation we observe acts at the transcriptional level, and not post-transcriptionally as envisaged by the omnigenic model[5,6].

How might our simple model give rise to complex regulatory networks? By analysing simulation trajectories, we noted that TUs lying near each other in 1D sequence space often joined the same cluster in 3D. As a result, the activity of these clustered beads is highly positively correlated. At the same time, cluster formation sequesters TFs and so reduces the likelihood that another cluster forms elsewhere. As a result, most long-range correlations are negative (Fig. 3A).

Crucially, these network properties depend on there being a low TF copy-number (as in vivo[23]) so TU beads do not become saturated. We therefore reasoned that increasing copy number should suppress correlations as more rarely transcribed TUs are pressed into use. Indeed, increasing $n$ reduces long-range negative correlations (Fig. 3Aiii,iv), and the fraction of nodes in the largest-connected component falls (Fig. 3B). Another way to think about this result is: if resources are plentiful, there is no need for sharing or competition, and all TUs can bind a TF independently of each other. If TFs do not switch and are permanently in the binding state (and $n = 10$), the network becomes even more highly connected (Fig. 3Ai).

**Modelling effect of mutations and SNPs in regulatory elements**. GWAS reveals that single-nucleotide polymorphisms (SNPs) in regulatory elements and TUs can lead to many small changes in transcriptional activity across the genome. To model

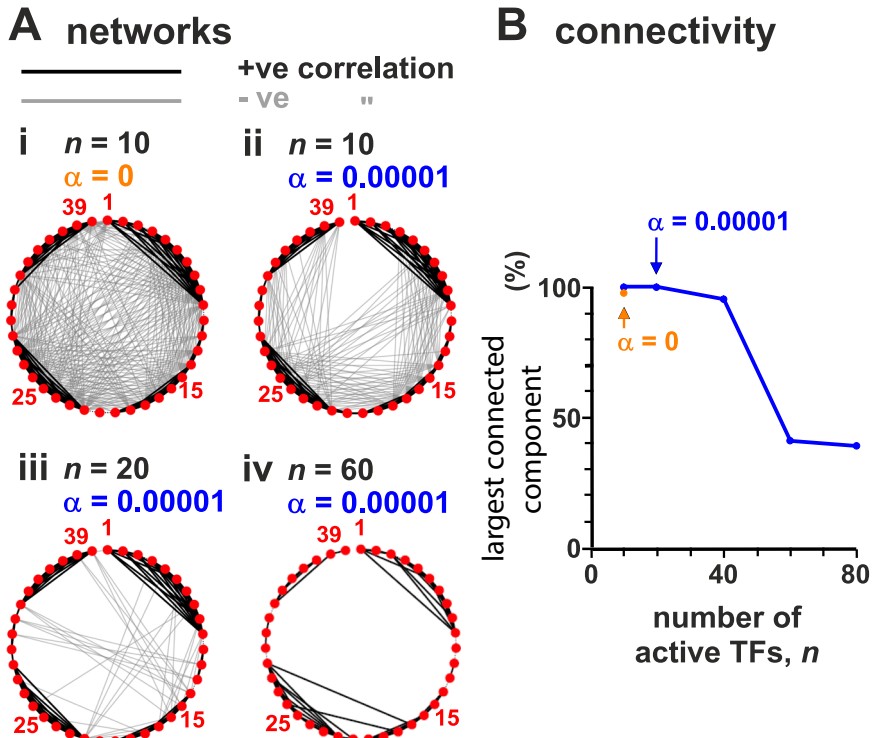

**Fig. 3 Regulatory networks formed by TU beads are percolating at low TF concentrations.** Simulations (as Fig. 1, with ≥800 simulations/condition) with different average numbers of active TFs (*n*) and switching rate (*α*). Networks were constructed by calculating the Pearson correlation between the transcription time series for all pairs of TUs; nodes represent each of the 39 TUs and edges are placed between nodes where there is a significant correlation (>0.15 in absolute value, corresponding to $p < 10^{-6}$; two-sided Student's *t*-test). **A** Effect of TF concentration and switching. Thirty-nine nodes are shown around the perimeter, and thick black and grey lines denote positive and negative correlations between transcriptional activities of bead pairs. **B** Effect of *n* on the fraction of nodes in the largest connected component.

this, we abrogate TF binding to one TU in the chain. Bead 930 is chosen first because it is usually highly active (Fig. 1C). This single "knock-out" affects in a statistically significant way the activity of almost half of the other TUs, both near and far away in sequence space (Fig. 4Aii). The immediately adjacent TU (i.e., bead 931) is down-regulated the most, while more distant ones are up-regulated (due to loss of a strong competitor). This knock-out also rewires the whole network, even though it still retains its small-world character (Fig. 4Aiii). Both positive and negative interactions are affected along the whole chain, as shown by a heat map of the change in Pearson correlation between TU transcriptional activities (Fig. 4Aiv).

We next systematically knock out each TU in turn. To quantify global effects, we define a "transcriptional difference" between the wild type and each knock-out based on a standard Euclidian-distance metric (SI, Supplementary Note 2); the larger this quantity, the more different the two states are. This difference varies >10-fold between different mutations (Fig. 4Bi).

Together, these observations are reminiscent of the behaviour of SNPs and eQTLs. Thus, each TU mutant can be seen as a SNP underlying an eQTL; then, those with low and high transcriptional differences (Fig. 4Bi,ii) are low- and high-effect eQTLs (low-effect mutants are often isolated in sequence space), and those with wide effects (e.g., bead 930 in Fig. 4A) may be viewed as omnigenic.

**Modelling loops, heterochromatin and euchromatin.** In mammalian genomes, promoter-enhancer pairs are often contained in loops stabilized by cohesin and the CCCTC-binding factor (CTCF)[42–44]. To investigate how such loops might affect

transcription, we incorporated eight permanent and non-overlapping loops at different positions in the chain (Fig. 5A, loops *a*–*h*). In reality, such loops may arise from extrusion by cohesin halted at convergent CTCF loops[42]. Our assumption of stable, permanent loops is quantitatively accurate in the limit in which the interaction between cohesin and CTCF is strong and long-lived. However, we expect the trends to be qualitatively similar for more transient loops consistent with the loop extrusion model as in refs. [19,43].

The inclusion of stable loops has subtle effects. For example, loop *h* encompasses three TUs (beads 905, 907, 930), and expression of one is slightly boosted compared to the unlooped case (Fig. 5B, C). This is consistent with the idea that looping switches on some genes during development[45], and can increase enhancer–promoter interactions[46,47]. However, up-regulation requires appropriate positioning of a TU within the loop. For instance, loop *d* encompasses two TUs (beads 396 and 404), and has no effect on their activity. Broadly speaking, looping up-regulates activity, but not invariably so, and—perhaps surprisingly—two of the three most up-regulated TUs (beads 33 and 886) are not contained in loops (Fig. 5C). Looping also extensively rewires the regulatory network (Fig. 5D, E). Globally, the increase in activity is modest, as incorporating all beads into closely packed loops only increases total activity by ~10%, with—once again—some TUs being down- as well as up-regulated (Fig. S3). This is consistent with experiments showing that the interplay between looping and expression is complex[48] but slight (e.g., knocking down human cohesin leaves expression of 87% genes unaffected, with global levels changing <30%[49]).

In simulations thus far, TFs bind strongly to TU beads, and weakly to all others to model binding to open euchromatin[19,50]. To

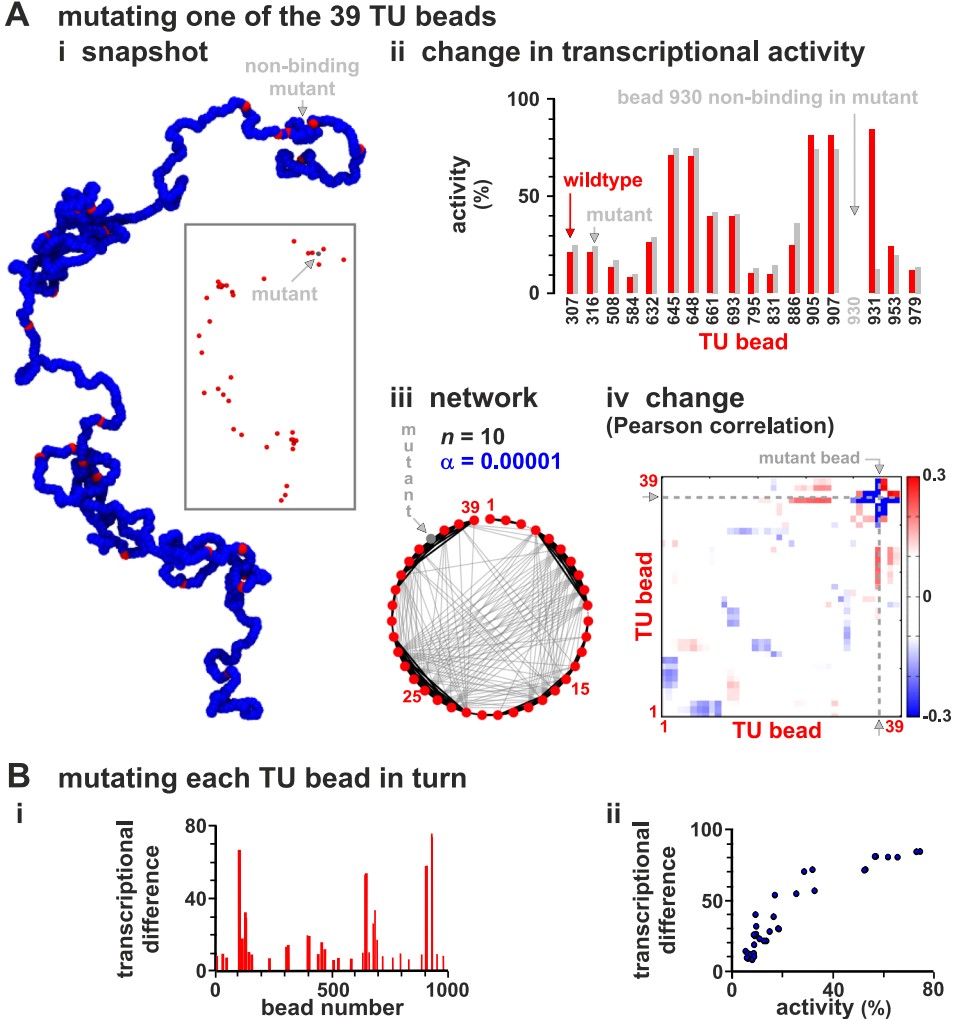

**Fig. 4 Modelling SNPs and eQTL action.** Sets of simulations (≥800 simulations/condition) where each of the 39 TU beads is made non-binding in turn (to represent 39 different SNPs in regulatory elements) are compared with those with the "wild-type" chain (as Fig. 1). **A** Chain with mutant (non-binding) TU bead 930. (i) Snapshot. TFs not shown (inset: same structure without blue beads). (ii) Transcriptional rates of the 17 TUs with significantly different values in mutant fibre compared with the wild-type one ($p \simeq 0.046$; two-sided Student's t-test). (iii) Regulatory network inferred from the matrix of Pearson correlations between activities of TUs. (iv) Change in Pearson correlation between TUs. **B** Results from simulations where each TU bead is mutated in turn, and the "transcriptional difference" from the wild type (see text and Supplementary Note 2) determined. (i) Transcriptional difference versus position along the chain. (ii) Positive correlation of transcriptional difference with TU activity in wild type. The plot shows that if we mutate a TU with high transcriptional activity, this leads to a larger difference.

investigate the effects of heterochromatin—which binds few TFs, carries few histone marks[51], and is gene poor and traditionally viewed as transcriptionally inert—we perform simulations where four of the most-active TUs (905, 907, 930, and 931) are embedded in a non-binding segment (running from bead 901–940). This has a dramatic effect (Fig. 6A–C): the activity of the TU beads now embedded in the non-binding island are at least halved, some nearby neighbors are down-regulated, and more distant ones up-regulated (again due to a reduction in competition; Fig. 6B, C). The regulatory network is also rewired (Fig. 5D, E).

Just as embedment in a non-binding segment down-regulates a TU bead, embedment in a weak-binding (euchromatic) one up-regulates it (Fig. S4). This shows our model effectively captures position effects where the local chromatin context strongly influences activity[52].

**Modelling a whole human chromosome.** We next model a whole mid-sized human chromosome (HSA 14, length 107 Mbp;

Fig. 7A) in a well-characterized and differentiated diploid cell (HUVEC, human umbilical vein endothelial cell). Now, multi-valent and switchable TFs (20% active at any moment) at a non-saturating concentration bind to a string with 35784 beads. As chromosome territories are often ellipsoidal, simulations are performed in an ellipsoid of appropriate size[7,53]; consequently, chromatin density is now higher than in simulations detailed above, with volume fractions comparable to those in vivo (~14%).

Chromatin beads are classified using DNase-hypersensitity data and ChIP-seq data for H3K27ac. DNase-hypersensitive sites (DHS) are excellent markers to locate promoters and enhancers (and so TF-binding sites[19,54]), whereas H3K27ac modifications strongly correlate with open chromatin[19]. Therefore, if the 3 kbp region corresponding to a chromatin bead has a DHS, then that bead is a TU; if it has H3K27ac, it is a euchromatin bead, and all other beads are non-binding (heterochromatic). We call this the "DHS" model. As properties of different chromatin segments have been catalogued using "hidden-Markov models" (HMMs)

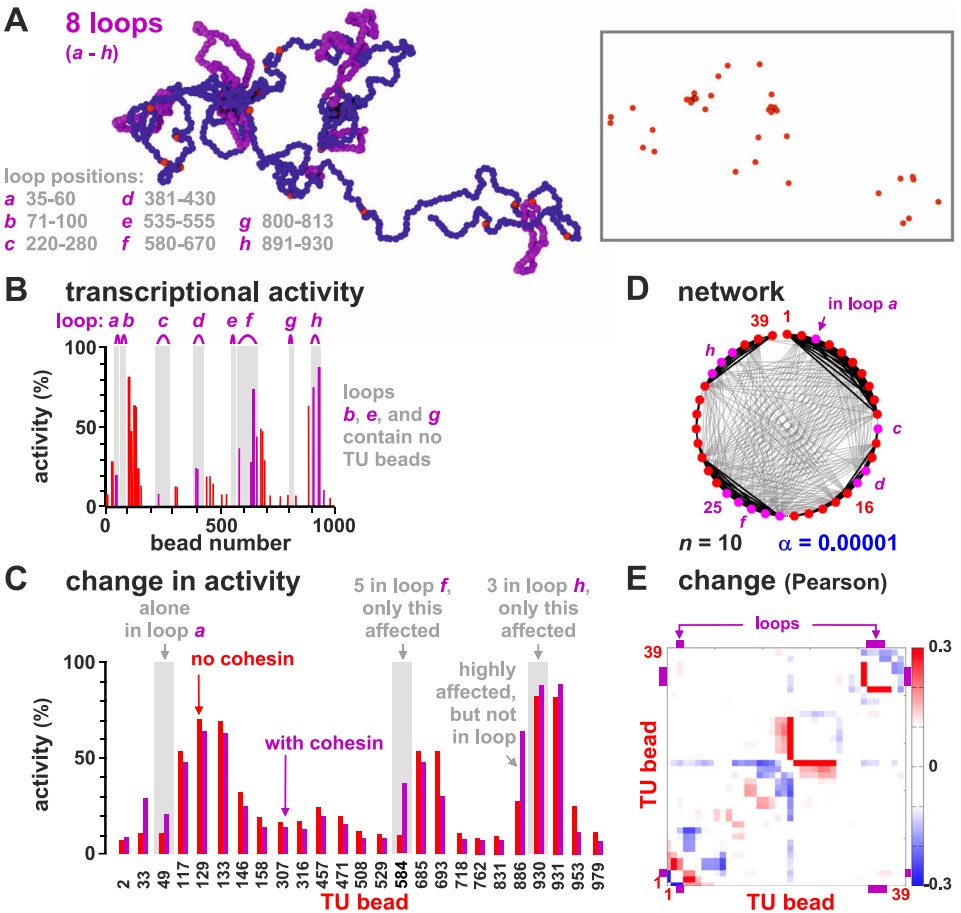

**Fig. 5 Looping subtly affects transcriptional activity.** Results of two sets of simulations are compared; one set as Fig. 1, in the other the chain contains eight permanent loops (to represent convergent loops stabilized by cohesin/CTCF). **A** Snapshot (beads within loops are magenta; TFs not shown; inset—same structure with only TUs shown). **B** Average transcriptional activity for each TU in the looped chain (magenta bars—values for TUs in loops; magenta arcs—loop positions). **C** Comparison between activity in wild type and looped configuration for the 25 TUs with significantly different values in the two sets ($p \simeq 0.003$; two-sided Student's $t$-test). **D** Regulatory network inferred from the matrix of Pearson correlations between expression levels of TUs (as Fig. 3A). **E** Change in Pearson correlation between TUs due to loops.

applied to many data sets[51], we alternatively classify beads according to HMM state; we call this the "HMM model" (Fig. S5). For more details, see Supplementary Note 3.

Simulations using the DHS model again yield clusters enriched in TUs and TFs (Fig. 7B). As before, aggregating data from many simulations allow determination of transcriptional activities of every bead, which we compare with those of corresponding regions determined experimentally[55] by GRO-seq (global run-on sequencing[56]); activities of all 3 kbp regions are ranked from high to low, binned into quintiles, and compared. In Fig. 7C, squares near the diagonal from bottom-left to top-right have high ranks (shown as red and yellow) compared to those off-diagonal (blue and purple) indicating good concordance between simulations and data. A specific sub-set of beads corresponding to SEs—which are highly active in vivo[57]—are also highly active in simulations (shown as white dots concentrated at top right). Plots showing the rank of transcriptional activities in simulations and experiments in selected genomic regions are shown in Fig. S6. Simulations yield patterns qualitatively closer to those obtained with GRO-seq than those given by poly(A)$^+$ RNA-seq, as the latter only include genic transcription. Concordance between results from simulations and GRO-seq is confirmed by the Spearman rank correlation (~0.38 for all beads; $p < 10^{-12}$; this measure is used because it is less sensitive to outliers; Fig. 7D). Restricting analysis just to TUs provides a more stringent

comparison (as all TUs bind TFs with equal affinity); it still yields a significant correlation ($r \simeq 0.32$, $p < 10^{-12}$; Fig. 7D). As neighbouring high-affinity regions tend to have roughly similar transcriptional rates in both simulations and data, we also average rates found in active "patches" (contiguous sets of beads which are either all TUs or all labelled as euchromatin), but found this has no significant effect (Fig. 7D). Concordance was confirmed using our HMM model (Fig. 7D, right, and Fig. S5). Adding cohesin-mediated looping to simulations involving the DHS model did not significantly change agreement with experimental data (e.g., for TUs only, $r \simeq 0.33$, $p < 10^{-12}$). Similar agreement with GRO-seq data was obtained from simulations applied to the H1 human embryonic stem-cell line (for TUs using the DHS model, $r \simeq 0.29$, $p < 10^{-12}$), and to the GM12878 cell line (DHS model, $r \simeq 0.33$, $p < 10^{-12}$).

As in the chromosome fragment simulations (Fig. S1B), the transcriptional activity of a TU in our model anticorrelates with the distance to the nearest TU. In our HSA14 simulations, the presence of heterochromatin slightly reduces the absolute value of the correlation, which however remains highly significant (Spearman correlation $r \simeq -0.83$, $p < 10^{-12}$). Interestingly, the experimental GRO-seq signal of a DHS also anticorrelates with the distance to the nearest DHS in a significant way, although more weakly than in simulations (Fig. S7; over the whole genome the Spearman correlation is $r \sim -0.23$, $p < 10^{-12}$).

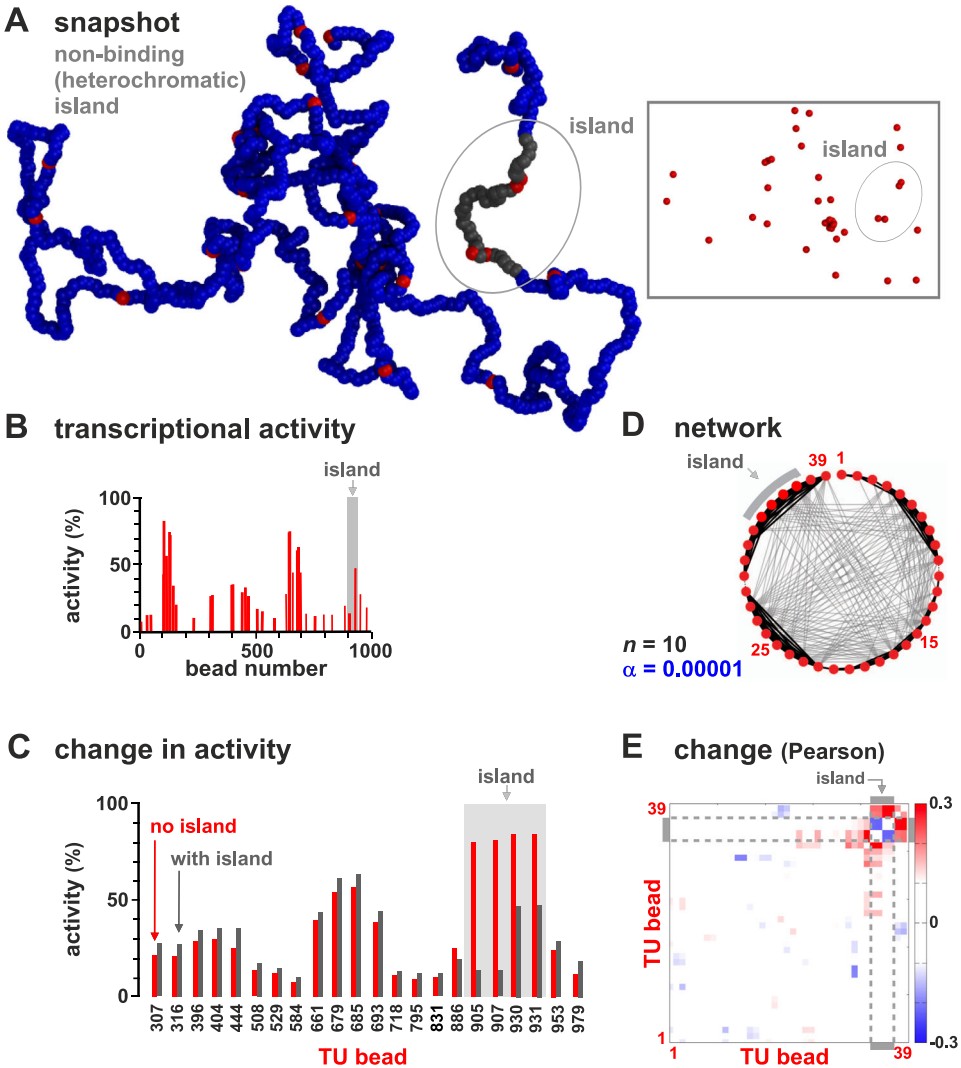

**Fig. 6 Neighboring heterochromatin affects transcriptional activity.** Results from two sets of simulations (at least 800 runs for each condition) are compared; one set as Fig. 1, in the other beads around TU beads 905, 907, 930, and 931 (from bead 901 to 940) are non-binding (to represent embedding the TU beads in heterochromatin). **A** Snapshot with heterochromatic beads shown in gray (TFs not shown; inset—the same structure with only TUs). **B** Average transcriptional activity for each TU. **C** Comparison of average transcriptional activity with respect to wild type for the 22 TUs with significantly different values in the two sets ($p \simeq 0.003$; two-sided Student's $t$-test). **D** Regulatory network inferred from the matrix of Pearson correlations between activities of TUs (as Fig. 3A). **E** Change in Pearson correlation between TUs due to heterochromatin.

**Networks inferred from simulations are qualitatively similar to experimental ones.** Regulatory networks emerging from our whole-chromosome simulations are again small-world and highly connected (Fig. S8 and Supplementary Note 4). To facilitate comparison with previous results, we select four segments of HSA14 that have the same length as the one considered in Fig. 3 (i.e., 3 Mbp), and roughly the same density of TUs; all four segments again have highly connected components (compare Fig. S8 and Fig. 3). However, patterns in real chromosomes and artificial fragments are quite different. In HSA14 networks, there are more positive interactions between sets of adjacent TUs and other sets that are >10 beads distant in sequence space (black lines across the middle of circles in Fig. S8).

Whole-chromosome networks also have the following statistical properties. First, their node-degree distribution decays exponentially (Fig. S9A)—as found in gene networks[58] but not in transcription factor interaction networks, which are often scale-free[59]. Second, they are modular (as clusters arising due to the bridging-induced attraction are the basic co-regulated

building blocks)—again as found in gene[58] and eQTL[60] networks. [Modularity is apparent from the blocks visible in the correlation matrices, such as in Fig. S2.] Third, node degree broadly correlates with transcriptional activity (Spearman correlation 0.59, $p$ value $< 10^{-12}$)—as in gene coregulation networks[58].

**Contact maps found by simulations are qualitatively similar to Hi-C.** We previously showed[16] that simulations involving two different TFs (binding to active and inactive regions, respectively) yield contact maps much like those found with Hi-C[42]. Therefore, we expected the present simulations to reflect Hi-C data poorly as they involve only one TF binding to the minor (i.e., active) fraction of the genome, so contacts made by this structured minority would be obscured by those due to the unstructured majority. Even so, simulations yield contact maps broadly similar to those obtained by Hi-C (Fig. 7E). To measure the agreement, we use a comparison based on contact maps restricted to TUs as anchors—which may be considered as equivalent to interactions obtained by promoter-capture HiC[61].

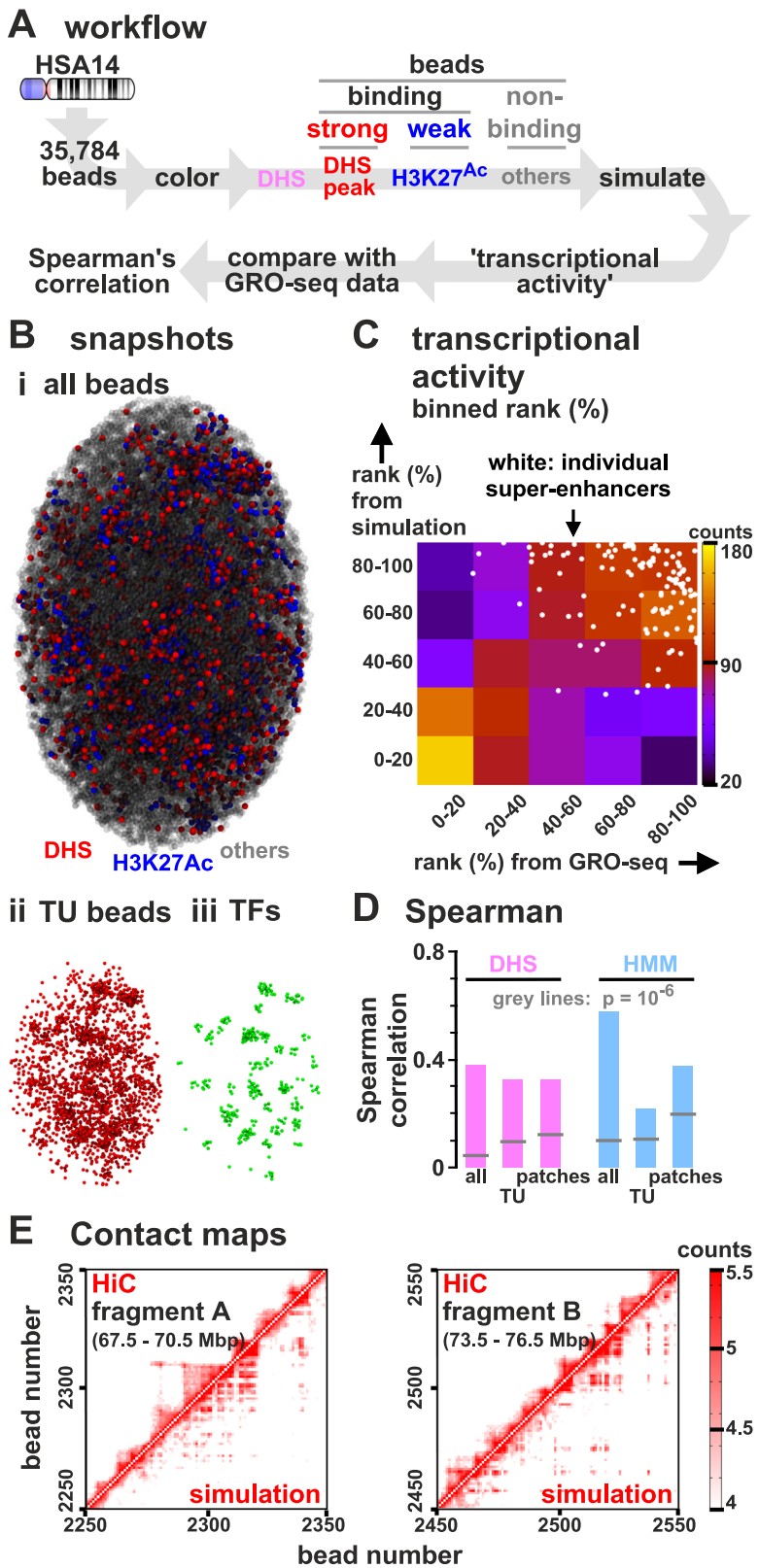

These yield good concordance (Fig. 7E; Pearson coefficient $r = 0.82$; $r = 0.47$ when monitoring only long-range contacts between TUs at least 300 kbp away, $p < 10^{-6}$ in both cases). The exponent with which contact probability decays with 1D distance is $\sim -1.1$ in experiments, and $\sim -0.8$ in simulations (fitted for 1D distances between $\sim 30$ kbp and 1.5 Mbp), both broadly consistent with the $-1$ value expected for a fractal globule[62]. The small discrepancy may point to our simulations slightly overestimating the weight of long-range contacts, perhaps because we do not include loop extrusion.

Overall the results obtained in our HSA14 simulations show that a simple model based on 3D chromatin organisation captures much of the complexity in 3D structure and transcription of a whole human chromosome.

**Fig. 7 Comparison of transcriptional activities of TUs on HSA14 in HUVECs determined using simulations and GRO-seq. A** Workflow (DHS model). Simulations (244 runs) involve a chain (35,784 beads) representing HSA14, and 1700 switchable TFs confined in an ellipsoidal territory. Beads are classified as TUs (red, strong binding), euchromatic (blue, weak binding), or heterochromatic (grey, non-binding). Transcriptional activities from simulations are compared with those of GRO-seq data, by measuring the Spearman rank correlation. **B** (i) Snapshot (TFs not shown). (ii, iii) TU beads and TFs in this configuration. **C** Comparison of transcriptional activities of TUs from simulations and GRO-seq (ranked from 0 to 100%, then binned in quintiles and showed as a heat map). A scatter plot of unbinned ranks of beads corresponding to SEs are superimposed (white circles). **D** Comparison of transcriptional activities from simulations (for both DHS and HMM models) and GRO-seq for all 3 kb regions/beads, only TUs, and only connected patches of binding beads (see text). All correlations are significant ($p < 10^{-6}$; two-sided Student's $t$-test, indicated by grey lines). **E** (i, ii) Capture-HiC-like contact maps obtained from simulations and experiments[42] showing logarithm of number of contacts between 30 kbp bins which contain TUs.

**Modelling chromosome 22 carrying the diGeorge deletion**. Our approach can, in principle, be applied to study any chromosome providing appropriate genomic data are available (e.g., on DNase hypersensitivity and histone acetylation). As a proof of principle, we studied the effect of deleting ~2.55 Mbp from HSA22—an alteration which is associated with the diGeorge syndrome (Fig. 8A) (https://dosage.clinicalgenome.org/clingen_region.cgi?id=ISCA-37446). This syndrome affects ~1 in 4000 people, and the variable symptoms include congenital heart problems, frequent infections, developmental delays, and learning problems.

We predict a multitude of small effects in TU activity, both near and far away from the deletion (see the Manhattan plot in Fig. 8Bi). In particular, most TUs are slightly up-regulated, as fewer TUs compete for the same number of factors, and the TUs which change the most have intermediate transcriptional activities in the wild type (Fig. S10). The $p$ values associated with the change in transcriptional activities vary widely, and comparison of the observed distribution with the null hypothesis (indicating that changes in measured transcription are due to random variation) shows the observed is highly enriched in small $p$ values (Fig. 8Bii), as is generally the case with results from GWAS[5,6]. The regulatory network is also re-wired (Fig. 8C). Results are consistent with measurements of differential gene expressions in patients, which showed both a large number of up-regulated and down-regulated genes[63]. A more quantitative comparison between experiments and simulations would benefit from having GRO-seq data that include non-genic transcription.

Clearly, this approach opens up a rich field of study. For instance, while there may be processes which occur in vivo which are not represented in our model, it could still give an indication of the genes most likely to be affected by any chromosome rearrangement.

## Discussion

We have described a parsimonious 3D stochastic model for transcriptional dynamics based on multivalent binding of factors and polymerases (TFs) to genic and non-genic transcriptional units (TUs) in a chain representing a chromatin fibre. A distinctive feature of our framework is that it is fitting-free, which means the model is truly predictive and can provide a mechanistic understanding of the phenomena we observe. On the other hand, the absence of fitting renders it challenging to obtain a fully quantitative agreeement between modelling and experiment.

In our simulations two types of fibres were considered: a 3 Mbp fragment with randomly-positioned TUs, which is useful to exemplify emerging trends, and human chromosomes 14 and 22 where TUs were appropriately positioned according to bioinformatic data. Despite deliberately excluding any explicit underlying network of biochemical regulation, our model nevertheless yields some notable results. These depend on having a low TF copy-number—a feature compatible with observations in vivo[23]. First, since TFs bind with the same affinity to all TUs, one might expect the latter to all be transcribed similarly, but they are not (Fig. 1). This is largely due to inter-TU spacing; TUs lying close together in 1D sequence space tend to be the most active (Fig. 1C) with positively correlated dynamics reminiscent of transcriptional bursting (Fig. 2B). This is because they often cluster into structures which are analogous to the phase-separated transcription hubs/factories seen experimentally[7,10], or to contact domains formed by accessible DNA sites found by high-resolution mapping of chromatin interactions by microC[30]. Second, switching off binding at any TU significantly affects the activity of many others, both near and far away in sequence space (Fig. 4). Third, introducing stable loops has subtle effects (Fig. 5), consistent with the result that cohesin knock-outs and degrons lead to small global changes in expression[49], although they can be important for inducible gene response in selected cases[46]. Fourth, transcriptional activity of a TU is strongly affected by the local environment in ways that are reminiscent of the silencing of a gene by incorporation into heterochromatin[52] (Fig. 6), or activation by embedment in euchromatin (Fig. S4). Fifth, the stochasticity seen in individual simulations reflects that detected by single-cell transcriptomics and single-cell Hi-C. Nevertheless, this variability does not prevent emergence of robust phenotypes in a cell population. Sixth, our simple fitting-free model predicts patterns of transcriptional activity in human chromosomes that promisingly and significantly correlate with experimental GRO-seq data (Fig. 7). This suggests that chromatin structure significantly constrains transcriptional activity. We hypothesise that additional downstream biochemical regulation, not included in our model, may provide a tool to adjust this underlying "structural" pattern of activity in a way which may be required for appropriate biological function.

Finally, our results enable us to reconcile two conflicting sets of data, namely that regulatory networks are both complex (as GWAS shows that thousands of loci around the genome control complex phenotypes[5,6]) and simple (as over-expressing just four Yamanaka factors switches cell fate[4]). Thus, our simulations reveal complex small-world networks of mutual up- and down-regulation (Figs. 3 and S8), consistent with GWAS results. However, increasing TF copy-number dramatically simplifies network structure (Fig. 3). We suggest such a simplification occurs when a fibroblast is reprogrammed into a pluripotent stem cell by over-expressing the Yamanaka factors; the high factor concentration simplifies the network so that the factors can combine to switch the phenotype (Fig. S11).

Taken together, these results suggest the activity—or inactivity—of every genomic region affects that of every other region to some extent. We describe our framework as "pan-genomic" (Fig. S11). This is reminiscent of the omnigenic model[5,6] in the sense that many loci are involved, all having small effects. However, it differs as it provides an underlying mechanism for pangenomic effects, by positing a direct and immediate effect of structure on regulation at the transcriptional level, which contrasts with the non-trivial post-transcriptional pathways envisioned by the omnigenic model. Additionally, our pangenomic model yields a natural framework to qualitatively understand mutually exclusive gene expression, when switching on one gene in a family turns off all others (as in

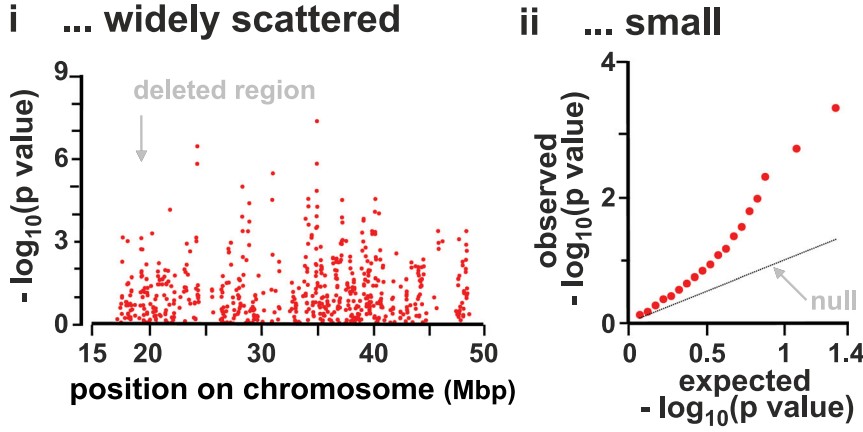

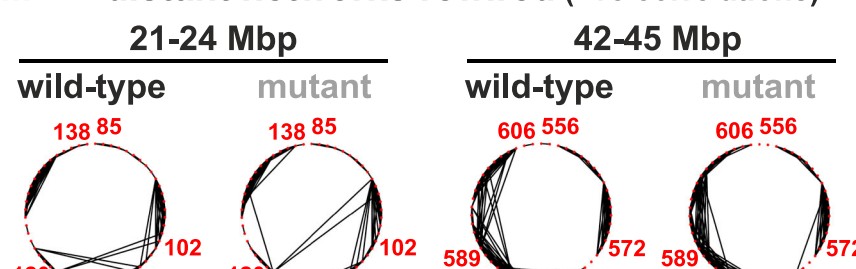

**Fig. 8 Modelling effects of the DiGeorge deletion in HSA22. A** Workflow. Simulations (800 simulations/condition) for wild type (17,102 beads) and deletion (16,250 beads, where wild-type beads 6305–7156 are cut, corresponding to a deletion of chr22:18,912,231–21,465,672 in hg19). [Agreement between predicted transcriptional activity and GRO-seq in HSA22 is similar to that found for HSA14 (here, Spearman correlation is $r \sim 0.29$, $p < 10^{-6}$; two-sided Student's $t$-test).] **B** (i) Manhattan plot showing $-\log_{10}(p\ \text{value})$ as a function of genomic position along HSA22 (position given in Mbp), for changes in TU transcriptional activities between wild type and deletion. (ii) Quantile–quantile plot showing expected versus observed values for $-\log_{10}(p\ \text{value})$ for the same data in (i). Expected values are computed from the normal distribution (these correspond to the null hypothesis according to which the change in transcriptional activities in the deletion is purely due to random variation). (iii) Regulatory networks of two 3 Mbp segments in chromosome 22 inferred from the Pearson correlation matrix. Edges show positive correlations >0.12 ($p = 0.0007$). Segments chosen have roughly the same number of nodes in 3 Mbp as the short fragment (Fig. 3Aii).

developing olfactory neurons[64]). The current model to explain this phenomenon postulates a coupling between *cis*-acting up-regulation and *trans*-acting down-regulation. The pangenomic networks we find provide exactly this type of regulatory interactions (Fig. 3). Our results are also consistent with recent experiments and mathematical models showing that subtle changes in 3D structure can lead to large changes in transcription[65,66]. On the other hand, it is challenging within our current model to account for local negative feedback mechanisms leading to noise reduction or oscillations[11], as these are more likely to arise biochemically (an example is the p53–Mdm2 system which achieves stabilisation of the cellular concentration of p53 via a negative feedback loop[67]).

In conclusion, we have developed a framework that can be applied to predict the transcriptional activity of any genomic fragment in health or disease (Figs. 7 and 8) providing

appropriate experimental data are available. Predictive power can be enhanced by incorporating additional TFs, and more suitable datasets of histone marks. Other features that can improve correlations between experiments and simulations are a more accurate modelling of cohesin loop formation by loop extrusion, and of the heteromorphic nature of chromatin[19]. We hope to report on work incorporating the latter two features in the future.

## Methods

**Polymer modelling.** We model chromatin fibres and chromosomes as bead-and-spring polymers. A fibre has $M$ monomers, each of size $\sigma$ (corresponding to 3 kbp, or 30 nm[24]), and $\mathbf{r}_i$ denotes the position of the $i$th monomer in 3D space. Multi-valent transcription factors (either active or inactive) are modelled as spheres, again with size $\sigma$ for simplicity. There are $n$ multivalent factors in a simulation (where $n$ is varied systematically, see text and "Results" section for details), and $N$ high-affinity binding sites, which we refer to as TU (or TU beads).

Any two monomers ($i$ and $j$) in the chromatin fibre interact purely repulsively, via a Weeks–Chandler–Anderson potential, given by

$$U_{\text{WCA}}^{ij} = 4k_{\text{B}}T\left[\left(\frac{\sigma}{r_{ij}}\right)^{12} - \left(\frac{\sigma}{r_{ij}}\right)^{6} + \frac{1}{4}\right] \tag{1}$$

if $r_{ij} < 2^{1/6}\sigma$ and 0 otherwise, where $r_{ij}$ is the separation of beads $i$ and $j$. There is also a finite extensible non-linear elastic (FENE) spring acting between consecutive beads in the chain to enforce chain connectivity. This is given by

$$U_{\text{FENE}}^{ij} = -\frac{K_f R_0^2}{2}\ln\left[1 - \left(\frac{r_{ij}}{R_0}\right)^2\right] \tag{2}$$

where $i$ and $j$ are neighbouring beads, $R_0 = 1.6\sigma$ is the maximum separation between the beads, and $K_f = 30k_{\text{B}}T/\sigma^2$ is the spring constant. With simulations including permanent cohesin loops (Fig. 7 in the main text, and Supplementary Fig. S4), neighbouring monomers and monomers forming loops interact via harmonic, rather than FENE springs,

$$U_{\text{harmonic}}^{ij} = K_{\text{h}}\left(r_{ij} - \bar{R}\right)^2 \tag{3}$$

where $i$ and $j$ are neighbouring beads, $K_{\text{h}} = 100k_{\text{B}}T/\sigma^2$ is the harmonic spring constant, and $\bar{R}$ is the equilibrium spring distance. For these simulations, we use $\bar{R} = 1.1\sigma$ for bonds joining neighbouring monomers along the chain, and $\bar{R} = 1.8\sigma$ for bonds joining loop-forming monomers. The harmonic potential is used instead of the FENE one to enhance numerical stability.

Finally, a triplet of neighbouring beads interact via a Kartky–Porod term to model the stiffness of the chromatin fibre. This term explicitly reads as follows:

$$U_{\text{KP}}^{ij} = \frac{k_{\text{B}}T\ell_{\text{p}}}{\sigma}\left[1 - \frac{\vec{t}_i \cdot \vec{t}_j}{|\vec{t}_i||\vec{t}_j|}\right] \tag{4}$$

where $i$ and $j$ are neighbouring beads, $\vec{t}_i$ is the tangent vector connecting beads $i$ to $i+1$, and $\ell_{\text{p}}$ is related to the persistent length of the chain: this parameter is set to $3\sigma$ in our simulation, which corresponds to a relatively flexible fibre—the resulting persistence length is within the range of values estimated for chromatin from experiments and computer simulations[68].

The interaction between a chromatin bead, $a$, and a multivalent TF, $b$, is modeled through a truncated and shifted Lennard–Jones potential, given by

$$U_{\text{LJ}}^{ab} = 4\epsilon_{ab}\left[\left(\frac{\sigma}{d_{ab}}\right)^{12} - \left(\frac{\sigma}{d_{ab}}\right)^{6} - \left(\frac{\sigma}{r_c}\right)^{12} + \left(\frac{\sigma}{r_c}\right)^{6}\right], \tag{5}$$

for $d_{ab}$ (the distance between the centres of chromatin bead and protein) smaller than $r_c$, and 0 otherwise. The parameter $r_c$ is the interaction cut-off; it is set to $r_c = 2^{1/6}\sigma$ for inactive proteins or for active proteins and non-binding chromatin beads (this cutoff results in a Weeks–Chandler–Anderson potential and purely repulsive interactions), or to $r_c = 1.8\sigma$ for an active protein and a binding chromatin bead (this results in an attractive interaction). In all cases, the potential is shifted to zero at the cut-off in order to have a smooth potential. Purely repulsive interactions are modeled by setting $\epsilon_{ab} = k_{\text{B}}T$, while attractive interactions are modeled using $\epsilon_{ab} = 3k_{\text{B}}T$ for active TF and low-affinity beads, and to $\epsilon_{ab} = 8k_{\text{B}}T$ for active TF and high-affinity (TU) beads.

A TU bead (or more generally any chromatin bead in Fig. 8D in the main text) is said to be transcribed if it is bound to a factor—i.e., if there is at least a TF whose centre lies within a range $r_c = 1.8\sigma$ away from the bead centre.

The time evolution of each bead in the simulation (whether TF or chromatin bead) is governed by the following Langevin equation:

$$m_i\frac{d^2\vec{r}_i}{dt^2} = -\nabla U_i - \gamma_i\frac{d\vec{r}_i}{d}t + \sqrt{2k_B T\gamma_i}\,\vec{\eta}_i(t), \tag{6}$$

where $U_i$ is the total potential experienced by bead $i$, $m_i \equiv m$ and $\gamma_i \equiv \gamma$ are its mass and friction coefficient (equal for all beads in our simulations), and $\vec{\eta}_i$ is a stochastic noise vector with the following mean and variance:

$$\langle\vec{\eta}(t)\rangle = 0; \quad \langle\eta_{i,\alpha}(t)\eta_{j,\beta}(t')\rangle = \delta_{ij}\delta_{\alpha\beta}\delta(t - t'), \tag{7}$$

where the Latin and Greek indices run over particles and Cartesian components, respectively, and $\delta$ denotes here the Kronecker delta.

As is customary[69], we set $m/\xi = \tau_{\text{LJ}} = \tau_{\text{B}}$, with the LJ time $\tau_{\text{LJ}} = \sigma\sqrt{m/\epsilon}$ and the Brownian time $\tau_{\text{B}} = \sigma^2/D$, where $\epsilon$ is the simulation energy unit, equal to $k_{\text{B}}T$, and $D = k_{\text{B}}T/\gamma$ is the diffusion coefficient of a bead of size $\sigma$. From the Stokes friction coefficient for spherical beads of diameter $\sigma$ we have that $\xi = 3\pi\eta_{\text{sol}}\sigma$ where $\eta_{\text{sol}}$ is the solution viscosity. One can map this to physical units by setting $T = 300$ K and $\sigma = 30$ nm, as above, and by setting the viscosity to the effective viscosity of the nucleoplasm, which is scale-dependent and ranges between 10 and100 cP for objects of the size of our chromatin bead[70]. This leads to $\tau_{\text{LJ}} = \tau_{\text{B}} = 3\pi\eta_{\text{sol}}\sigma^3/\epsilon \simeq 0.6$–6 ms. The Brownian time $\tau_{\text{B}}$ is our unit of time in simulations. The numerical integration of Eq. (6) is performed using a standard velocity-Verlet algorithm with time step $\Delta t = 0.01\tau_{\text{B}}$ and is implemented in the LAMMPS engine[71]. Protein switching is including by

stochastically changing the type of TF beads every 10,000 timesteps (equivalently, every 100 Brownian times), with probabilities such that the switching off rate is of $\alpha = 10^{-5}\tau_{\text{B}}^{-1}$, or 0.017–0.17 s$^{-1}$. In simulations of the toy model (Figs. 1–7 in the main text and Suppl. Figs. S1–S4), the switching on rate is equal to $\alpha$; in chromosome 14/22 simulations (Fig. 8 in the main text and Suppl. Fig. S5), it is equal to $\alpha/4$. Consequently, in steady state the average number of active and inactive proteins is equal in simulations of the toy model, whereas the average number of inactive proteins is fourfold larger than that of active proteins in chromosome 14/22 simulations.

For more details on simulations, see Supplementary Notes 1 and 3.

**Reporting summary**. Further information on research design is available in the Nature Research Reporting Summary linked to this article.

## Data availability
The datasets generated during and/or analysed during the current study have been deposited in Edinburgh DataShare [https://doi.org/10.7488/ds/3110]. To compare the predicted transcriptional activity of chromosome 14 outputted by our simulations with experiments, we use GRO-seq data. For HUVECs, we use the datasets GEO: GSM2486801, GSM2486802, GSM2486803. For hESCs, we use GEO: GSM1579367, GSM1579368. Super-enhancer regions considered here are those identified in ref. [57], and available in the dbSUPER database [http://asntech.org/dbsuper/].

## Code availability
The code used for the simulation is LAMMPS, which is publicly available at https://lammps.sandia.gov/. Custom codes written to analyse data are available from the corresponding author upon request, or they can be downloaded from https://git.ecdf.ed.ac.uk/dmarendu/omnigenomic-model (access can be requested from the corresponding author).

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

## Acknowledgements

We thank the European Research Council (ERC CoG 648050 THREEDCELLPHYSICS) for support.

## Author contributions

C.A.B., N.G., D. Michieletto, A.P., P.R.C., M.C.F.P., and D. Marenduzzo designed research; C.A.B., M.C.F.P., and D. Marenduzzo performed research; C.A.B., N.G., D. Michieletto, A.P., M.C.F.P., P.R.C., and D. Marenduzzo analysed the data and wrote the manuscript.

## Competing interests

The authors declare no competing interests.
