## [Peer Review File · Nature Communications]

Complex small-world regulatory networks emerge from the 3D organisation of the human genomeEditorial Note: This manuscript has been previously reviewed at another journal that is not operating a transparent peer review scheme. This document only contains reviewer comments and rebuttal letters for versions considered at *Nature Communications*.

REVIEWERS' COMMENTS

Reviewer #2 (Remarks to the Author):

The authors have addressed most of my comments. I believe that it puts forward a thoughtful and provoking idea that deserves to be published and discussed by the broader community.

Reviewer #3 (Remarks to the Author):

I thank the authors for their careful and helpful replies to my comments.

I'm not sure if my comment regarding the selective effect of cohesin on gene induction has come through clearly. I was mainly referring to the findings of Cuartero et al., *Nat Immunol* 2018 (<https://pubmed.ncbi.nlm.nih.gov/30127433/>). It might be interesting for the authors to discuss this paper in the context of their model.

Another two very recent papers - in fact, still preprints - that the authors may wish to discuss are kinetic models of the effects of distance on enhancer-promoter communication that propose a more subtle relationship between enhancer-promoter contacts and transcriptional bursting: Xiao et al. (<https://www.biorxiv.org/content/10.1101/2020.10.22.351395v1>) and Zuin et al. (<https://www.biorxiv.org/content/10.1101/2021.04.22.440891v1>).

As before, I defer to the other two reviewers as regards the technical validity and advance of this work.

Reply to Referee 2

COMMENT:

The authors have addressed most of my comments. I believe that it puts forward a thoughtful and provoking idea that deserves to be published and discussed by the broader community.

RESPONSE:

We are grateful to the Reviewer for her/his positive view on our revised version and for recommending publication in Nature Communications.

Reply to Reviewer 3

COMMENT:

I thank the authors for their careful and helpful replies to my comments.

I'm not sure if my comment regarding the selective effect of cohesin on gene induction has come through clearly. I was mainly referring to the findings of Cuartero et al., Nat Immunol 2018 (<https://pubmed.ncbi.nlm.nih.gov/30127433/>). It might be interesting for the authors to discuss this paper in the context of their model.

RESPONSE:

We thank the reviewer for her/his positive view on our revision.

Regarding the comment on cohesin effects, we have now added a brief discussion of the paper by Cuartero et al. in our final version.

COMMENT:

Another two very recent papers - in fact, still preprints - that the authors may wish to discuss are kinetic models of the effects of distance on enhancer-promoter communication that propose a more subtle relationship between enhancer-promoter contacts and transcriptional bursting: Xiao et al. (<https://www.biorxiv.org/content/10.1101/2020.10.22.351395v1>) and Zuin et al. (<https://www.biorxiv.org/content/10.1101/2021.04.22.440891v1>).

RESPONSE:

We have also briefly discussed these two papers in the final version.